# Origami-based tunable truss structures for non-volatile mechanical memory operation

Hiromi Yasuda[1], Tomohiro Tachi[2], Mia Lee[1] & Jinkyu Yang[1]

Origami has recently received significant interest from the scientific community as a method for designing building blocks to construct metamaterials. However, the primary focus has been placed on their kinematic applications by leveraging the compactness and auxeticity of planar origami platforms. Here, we present volumetric origami cells—specifically triangulated cylindrical origami (TCO)—with tunable stability and stiffness, and demonstrate their feasibility as non-volatile mechanical memory storage devices. We show that a pair of TCO cells can develop a double-well potential to store bit information. What makes this origami-based approach more appealing is the realization of two-bit mechanical memory, in which two pairs of TCO cells are interconnected and one pair acts as a control for the other pair. By assembling TCO-based truss structures, we experimentally verify the tunable nature of the TCO units and demonstrate the operation of purely mechanical one- and two-bit memory storage prototypes.

[1] Department of Aeronautics & Astronautics, University of Washington, Seattle, WA 98195-2400, USA. [2] Graduate School of Arts and Sciences, University of Tokyo, Tokyo 153-8902, Japan. Correspondence and requests for materials should be addressed to J.Y. (email: jkyang@aa.washington.edu)

Mechanical memory operations can be highly useful not only to mimic electronic/optical memory devices, but also to store the flow of mechanical energy for sound isolation, heat insulation, and energy harvesting purposes[1–5]. These mechanical devices can function in harsh environments, such as space and nuclear power plants, where extreme thermal, mechanical, and radiation conditions can hinder the operation of electronic devices. The robustness of mechanical systems, together with nanoelectromechanical technologies, has indicated the possibility and effectiveness of mechanical memory storage and computing devices[6, 7]. In previous studies, however, the operations of mechanical memory devices are mostly limited to an individual one-bit memory level (few attempts on multi-bit memories[8]), and the possibility of interconnected operations across the neighboring bits has not been fully explored.

Here, we study how we can realize a two-bit mechanical memory operation by using origami cells. These origami units can work in a modular way, and they can interact with each other to demonstrate hierarchical, multi-bit memory operations. To achieve this, we exploit the tunability of origami cells, which enables the coupling and bit-flipping behavior between the adjacent cells. We show that origami-based structures provide an excellent platform to manipulate their tunable mechanical characteristics, such as stability and stiffness, in a controllable manner.

Origami has been a popular method for designing building blocks to construct mechanical metamaterials[9–17]. In particular, a quadrangular mesh origami, e.g., Miura-ori pattern[18], has been studied extensively, because it offers a single degree of freedom (DOF) mechanism of folding without relying on the elasticity of materials. This structure is called rigid (foldable) origami, and its 1-DOF motion can be beneficial for the control of deployable planar structures, such as solar panels and sails[19, 20] and sandwich core materials[21].

In contrast to the rigid planar origami, volumetric origami generally inherits a highly nonlinear elastic behavior, at the sacrifice of non-rigid deformation of panels. The coupled behavior of folding and deformation can result in versatile kinematic and dynamic motions. However, this multi-DOF behavior with deformable surfaces also poses formidable challenges in the analysis of volumetric origami. We here investigate the mechanics of volumetric origami, specifically triangulated cylindrical origami (TCO)[22–28], which can develop coupled dynamics of axial and rotational motions during folding (Fig. 1a). We demonstrate that the behavior of this TCO can be predicted analytically by modeling its deformable surfaces into

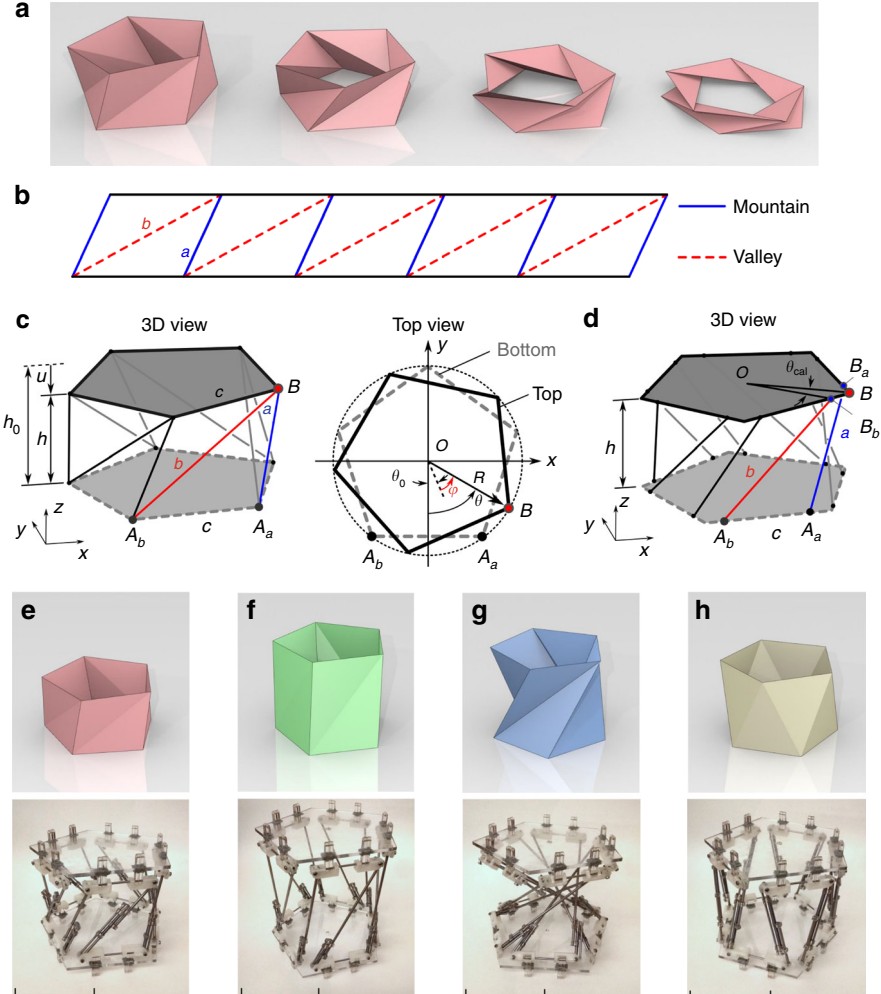

**Fig. 1** Geometry of triangulated cylindrical origami (TCO). **a** Folding motion of the TCO. **b** The flat sheet with crease patterns consisting of mountain crease lines (*a* shown as *blue solid lines*) and valley crease lines (*b* shown as *red dashed lines*). **c** Truss version of the TCO, where all facets are removed and crease lines are replaced by linear springs with a spring constant of *k*. **d** Modified truss structure for the fabrication of physical prototypes. **e–h** Graphical illustrations of four different TCO configurations (upper row) and digital images of their physical prototypes (lower row) (scale bar 100 mm). Their initial configurations are $(h_0, \theta_0) = (90 \text{ mm}, 46°)$, $(150 \text{ mm}, 40°)$, $(140 \text{ mm}, 92°)$, and $(119 \text{ mm}, 0°)$ from left to right

the network of truss elements and by applying the minimum potential energy principle. We find a rich tunability in this TCO structure, which enables the design of monostable/bistable, zero-stiffness, and bifurcation structures from one-parameter family of the initial geometry. By assembling multiple origami-based truss structures, we validate the tunable nature of the TCO units, and furthermore demonstrate the feasibility of mechanical memory storage units with non-volatile, bit-flipping behavior.

## Results

**Geometry of the triangulated cylindrical origami.** The TCO consists of repeating triangular arrays, which are characterized by valley crease lines (length $a$) and mountain crease lines (length $b$) as shown in Fig. 1b. Top and bottom surfaces of the TCO unit cell are $n$-sided polygons (e.g., $n = 5$ in Fig. 1) with side length $c$. Since the TCO is not a rigid foldable origami, folding/unfolding motions cause the warping of each facet, which may result in surface fatigue and damage under repeated usage. To overcome this issue while preserving the key characteristics of the TCO, we replace its surfaces with purely elastic truss members, which support tension/compression by using linear springs (Fig. 1c; Supplementary Movie 1 for the comparison between the paper- and truss-based TCO models). If we assume that the top and bottom surfaces always share the same rotational axis during folding/unfolding, we can characterize the shape of the unit cell by defining its height ($h$), relative angle between the top and bottom polygons ($\theta$), and radius of the circle circumscribing the

polygon ($R$). Note that for the sake of mathematical simplicity, $\theta$ is defined as an angle between $OB$ and the perpendicular bisector of $A_aA_b$ as shown in the top view of Fig. 1c. Letting $h_0$ and $\theta_0$ be the initial height and relative angle respectively, we can express deformations of the structure by axial displacement $u = -(h - h_0)$ where compression is defined to be positive, and rotational angle $\varphi = \theta - \theta_0$ (see Materials and Methods, and Supplementary Note 1 for more details on the modeling of the TCO).

In this truss model, two crease lines $a$ and $b$ are intersecting at a vertex of the polygon (e.g., $B$ in Fig. 1c). For the fabrication of this truss model, we need to secure space for mechanical joints. Thus, we modify the geometry of the TCO model, such that the two crease lines avoid intersecting (Fig. 1d). The difference between the original (Fig. 1c) and modified (Fig. 1d) models is characterized by the correction of the relative angle ($\theta_{cal}$ in Fig. 1d, see Supplementary Fig. 1). By adopting this modified model, we fabricate, test, and analyze four different types of the TCO structures in various combinations of $h_0$ and $\theta_0$. Figure 1e–h shows the graphical illustration of these four original models (top row) and the digital images of their modified physical prototypes (bottom row): $(h_0, \theta_0) = (90 \text{ mm}, 46°)$, $(150 \text{ mm}, 40°)$, $(140 \text{ mm}, 92°)$, and $(119 \text{ mm}, 0°)$. In these models, we use $R = 90 \text{ mm}$ and $\theta_{cal} = 9.7°$. See Materials and Methods, and Supplementary Note 2 for more details on the experimental configuration.

**Compression test on single unit cells.** To understand the folding behavior of the TCO-based structure, we calculate

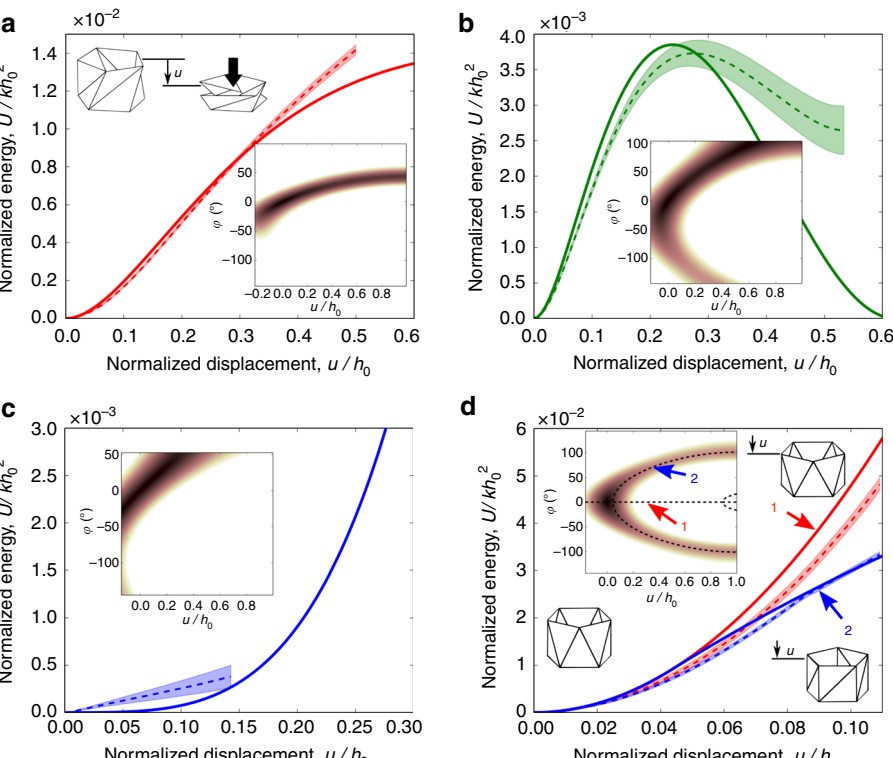

**Fig. 2** Folding behaviors of the TCO cells. **a–d** The energy analysis for the TCO-based truss structures shows remarkably different behaviors: **a** Monostability at $(h_0, \theta_0) = (90 \text{ mm}, 46°)$; **b** bistability at $(150 \text{ mm}, 40°)$; **c** zero-stiffness mode at $(140 \text{ mm}, 92°)$; and **d** bifurcation at $(119 \text{ mm}, 0°)$. The displacement is normalized by $h_0$, and energy is normalized by $kh_0^2$. Experimental results (mean value is shown as *dashed curves*, and s.d. is represented by *colored areas*) show qualitative agreements with the analytical predictions (*solid curves*). The inset plots show the equi-potential plots of $U/kh_0^2$ as a function of $u/h_0$ and $\varphi$, in which highlighted trajectories indicate the valley of minimum potential energy. In the experimental curves, the range of $u/h_0$ is restricted by the folding motions of the TCO-based truss prototypes (Supplemental Movie 3). For example, the highly twisted shape of the zero-stiffness TCO prototype (Fig. 1g) causes the truss elements overlap in the early stage of folding, allowing only ~15% of $u/h_0$ as shown in the panel **c**. The moderately twisted geometry of the monostable and bistable cases (Fig. 1e, f) permit more compression, allowing ~ 50% folding of the truss structure in terms of $u/h_0$ as shown in **a**, **b**

the total elastic energy ($U$) stored in the TCO cell as a function of $u$ and $\varphi$ (Materials and Methods; Supplementary Notes 1 and 2). The insets of Fig. 2a–d shows the surface maps of $U$ for the four models, where dark colored region indicates the valley of the minimum energy level. This highlighted region forms a near-curve trajectory in this configuration space, indicating that the TCO-based structure exhibits a mechanism of pseudo 1-DOF. Thus, simultaneously compressive and rotational motions of the TCO will follow this trajectory to satisfy the minimum potential energy principle (Materials and Methods). We can also calculate the change of the normalized energy ($U/kh_0^2$ where $k$ is the elastic constant of the linear truss element) under non-dimensionalized axial compression ($u/h_0$) by imposing $\partial U/\partial \varphi = 0$ (see Supplementary Note 3 and Supplementary Movie 2 for this uni-axial test). The solid curves in Fig. 2 represent analytical results predicted by the minimum potential energy trajectory in the inset surface maps. The experimental measurements with s.d. are denoted by dashed curves with bands. Note that in experiments, the range of $u/h_0$ is restricted by the folding motions of the TCO-based truss prototypes (Supplementary Movie 3). Within the measurement range, the experimental data corroborate these analytical results.

Comparing the four plots in Fig. 2, we observe remarkably different trends: monostable, bistable, zero-stiffness, and bifurcation behaviors, respectively. If $(h_0, \theta_0) = (90\ mm, 46°)$, the structure possesses only one minimum energy state at $u = 0$ (Fig. 2a). Therefore, the total energy increases monotonically as the TCO cell is compressed, implying a monostable property. If $(h_0, \theta_0) = (150\ mm, 40°)$, there exist two local minimum states along the energy valley as shown in Fig. 2b, indicating bistability. The TCO-based structure can also exhibit zero tangential stiffness, so called zero-stiffness mode, in which the application of axial compression does not create significant axial force or torque at the initial stage. Therefore, the total energy increases at an extremely low rate around $u = 0$ (Fig. 2c). The discrepancy between the analytical and experimental results may be attributed to the dissipative factors, including the friction of the mechanical joints in the truss elements. Nonetheless, we observe much smaller stiffness in this model compared to the previous two cases (0.26% and 0.18% in terms of the linearized initial stiffness relative to those of the monostable and bistable cases, respectively; see Supplementary Fig. 3). We analytically find that this zero-stiffness mode can be obtained when $\theta_0 = \pi/2$. Interestingly, this mode is independent of $k$ and $h_0$ (mathematical proof in Supplementary Note 4). This zero-stiffness mode can be potentially useful for impact absorption applications of origami, while maintaining its reusable and tailorable feature.

Last, we observe that the TCO-based truss can experience bifurcation if $\theta_0 = 0$ (Fig. 2d). That is, in the initial stage of the folding, the TCO-based structure is axially compressed without rotation. However, if it reaches a bifurcation point, there are three branches: one unstable branch (continuing pure compression without rotation as indicated by arrow 1 in Fig. 2d) and two stable branches (starting to develop twisting motions in one or the other direction as pointed by arrow 2 in Fig. 2d). The two different trends in the uni-axial testing verify this pitchfork bifurcation behavior (Fig. 2d and Supplementary Fig. 4; see Supplementary Note 5 and Supplementary Movie 4 for the specially devised uni-axial compression setup). Overall, the results from these four prototypes manifest versatile dynamics of the TCO, which can be controlled simply by altering its initial geometry (i.e., $h_0$ and $\theta_0$, more details in Supplementary Note 6; Supplementary Fig. 5).

**Mechanical memory device (one-bit memory operation).** Using this TCO-based truss structure as a unit cell, we further investigate the folding mechanism of multi-cell structures composed of

serially stacked TCO cells. We start with a two-cell structure that consists of identical monostable TCO units with $(h_0, \theta_0) = (90\ mm, 46°)$. They are linked together by sharing the interfacial polygon (Fig. 3a). Note that the chirality of the TCO cells is important in the multi-cell architectures. In this two-cell level, we arrange the cells in the opposite chirality, i.e., $(h_0, \theta_0) = (90\ mm, \pm 46°)$, such that they collectively show an interesting coupling motion. To test the dynamics of the combined structure, we fix the right end of the stacked prototype to the wall and impose pre-compression $u_C = 45\ mm$ to the left end of the system (Fig. 3a; Supplementary Fig. 6). Then, the total elastic energy of the system will differ depending on the rotational angles of the two unit cells, characterized by $\varphi_1$ and $\varphi_2$. Note that these angles are measured with respect to the initial positions of the left and central polygons, respectively. The inset of Fig. 3b shows the analytical values of $U$ as a function of $\varphi_1$ and $\varphi_2$, where the highlighted zone represents the valley of the minimum potential energy. We find that the pair of TCO cells collectively possess two local minimum states: one with the right cell folded and the other with the left cell folded (see the graphical illustrations in the inset of Fig. 3b).

The normalized elastic energy can be re-plotted as a function of $\varphi_1$ by imposing $\partial U/\partial \varphi_2 = 0$. Figure 3b evidently shows a symmetric double-well potential. This demonstrates that a pair of monostable TCO cells can successfully form a bistable system, requiring energy to overcome the potential barrier for the transition between the two stable states. Note that this potential barrier can be manipulated by controlling precompression, implying that the system features tunable potential barrier. Let '1' be the state where the first unit cell is folded, and '0' be the

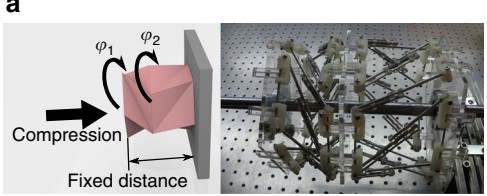

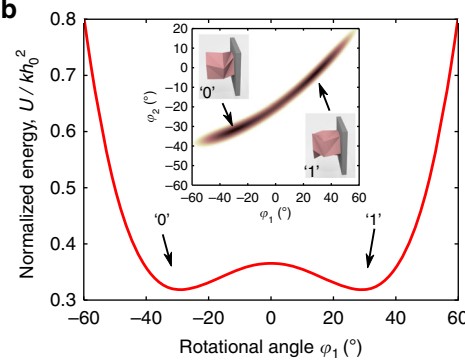

**Fig. 3** A pair of TCO cells' capability to demonstrate mechanical memory storage. **a** Two monostable TCO-based unit cells are connected horizontally. We fix the distance between leftmost and rightmost polygons by imposing a constant distance between them. Photograph of the corresponding configuration is shown in the right panel. **b** The normalized elastic energy as a function of $\varphi_1$ shows the double-well potential numerically. The inset shows the surface map of the elastic energy as a function of both $\varphi_1$ and $\varphi_2$, where the highlighted region denotes the valley of the map corresponding to the minimum potential energy trajectory. There exist two minimum states, and the illustrations show the schematic shapes of the pair of TCO cells at these points. Here, the configuration of the folded right cell represents '0', while the one with the folded left cell denotes '1'

state where the second unit cell is folded. Then we can use this system as a TCO-based mechanical memory device, which can store bit information ('1' or '0') by exploiting the double-well potential. One advantage of this mechanical memory is its non-volatility, meaning that it can store bit information stably without the necessity of external residual torque. To change the states from '0' to '1' or vice versa, we control only $\varphi_1$ so that the two-unit cell system can switch its state (see Supplementary Note 7 and Supplementary Movie 5 for experimental verification).

**Two-bit memory operation.** Now we demonstrate a two-bit memory operation. We use two pairs of the TCO cells, i.e., four identical units of the TCO-based truss elements ($h_0 = 90$ mm) in the sequence of $\theta_0 = [46°, −46°, 46°, −46°]$. Similar to the previous setup for the single bit operation, we fix the rightmost polygon to the wall in both translational and rotational directions, while we let the other polygons rotate freely. We apply precompression of $u_C = 50$ mm and 47.5 mm to the first and second pairs respectively, such that the two pairs maintain the specified compressed states throughout the operation. Here, we intentionally introduce distinctive $u_C$ values to break symmetry, thereby inducing controlled coupling behavior between the two bits (further details to be explained later).

We first analyze the total elastic energy of the system in relation to the deformed status of the two single-bit memory units. We represent the deformation of these two pairs by measuring the rotational angles $\varphi_1$ and $\varphi_3$, which measure the twisted angles of the first and third polygons respectively with

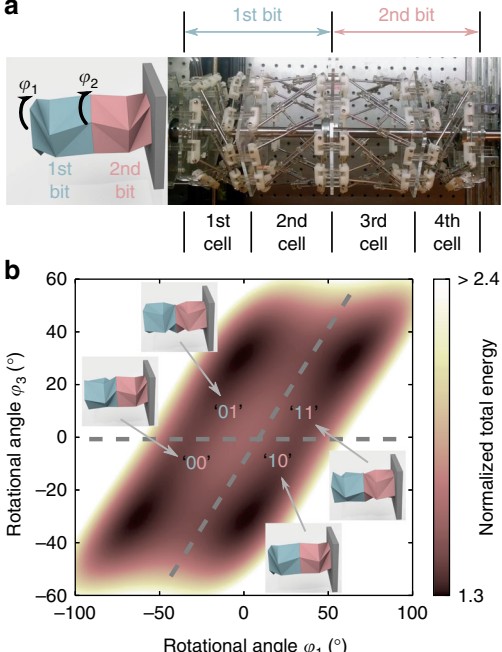

**Fig. 4** Two pairs of TCO cells' to construct two-bit memory. **a** Two-bit memory is composed of four TCO-based unit cells as shown in a schematic illustration (*left*) and photograph (*right*). The first two units from the *left* (i.e., *left* pair in *blue color*) form a first bit, and the other two unit cells (i.e., *right* pair in *red color*) form a second bit. The states of these two bits are determined by the rotational angles of the first and third unit cells, $\varphi_1$ and $\varphi_3$. **b** The potential energy of the system as a function of $\varphi_1$ and $\varphi_3$ is analyzed, where a darker region represents a lower level of the normalized potential energy (color map). We divide the $\varphi_1$ and $\varphi_3$ space into four regions: '00', '01', '10', and '11'. Inset illustrations indicate a representative shape of the TCO cells for each region

respect to their uncompressed positions (Fig. 4a). The analytical results are shown in Fig. 4b, where we identify four minimum energy states representing '00', '01', '10', and '11'. Here the first and second numbers indicate the first bit (left origami pair denoted in blue color in the inset of Fig. 4b) and second bit (right origami pair in red color), respectively. For example, '10' is the state where the first bit shows '1', and the second bit shows '0', following the definition of on and off status from the memory operation as illustrated in Fig. 3b.

For reading the memory state, we measure the rotational angles $\varphi_1$ and $\varphi_3$ individually by using a pair of non-contact laser Doppler vibrometers (Supplementary Note 7). We note in passing here that a mechanical approach of memory readout is also possible by measuring the torsional stiffness of the origami system. Similarly, the frequency response of the system can be also recorded to predict the stiffness of the system and thereby to read its memory state. Further details are described in Supplementary Note 8.

The next step is to test the operation of the two-bit memory. Unlike the operations of conventional memories (i.e., manipulation of each bit one by one), we demonstrate a unique operation of the two-bit memory by utilizing controlled coupling behavior between the two bits. In particular, this operation flips the second bit if the first bit is '1', which indicates that the first bit can control the state of the second bit. In this process, however, the second bit does not affect the state of the first bit, thus exhibiting a one-directional coupling mechanism. We demonstrate this operation by applying a pulse input to the first bit and measuring the response from the second bit. Specifically, we impose a trapezoid-shaped waveform on $\varphi_1$ to systematically change $\varphi_1$, which results in the alternation of the first bit between '1' and '0' (see Supplementary Note 9 and also ref. [29] for the details of a pulse operation technique).

The key point here is to verify that the onset of the control pulse (i.e., '1') can flip the information stored in the second bit. This process can be represented by the following two cases: '00' → '11' and '01' → '10'. The former corresponds to the conversion of the second bit from off to on (i.e, '0' to '1'), while the latter implies the opposite case that the second bit changes from on to off (i.e, '1' to '0') as the first bit is turned on.

We start with demonstrating the first case ('00' → '11'). Figure 5a illustrates the conceptual chart of the sequential operation of $\varphi_1$ and the consequential change of $\varphi_3$. The corresponding evolution of the TCO pairs' states is plotted in Fig. 5b, where the red curve denotes the experimental result of $\varphi_1$ and $\varphi_3$. The system is initially positioned at '00', as denoted by the state at (i) in Fig. 5a, b. As we apply the pulse input to the first bit (i.e., $\varphi_1 = −62°$ to 62°, see Supplementary Fig. 9c in the Supplementary Note 9 for the detailed pulse shape), the first bit changes its state from '0' to '1' in the beginning of the operation. See the transition of the experimental curve from point (i) to point (ii) in Fig. 5b. This onset of the first bit eventually flips the second bit from '0' to '1' (i.e., $\varphi_3 = −31°$ to 28°, see the state (iii) in Fig. 5b and Supplementary Movie 6 and Supplementary Note 9). Thus, we verify the transition from the initial state '00' to the final state '11' without resorting to any direct excitation applied to the second bit.

Next, we move on to demonstrate the second case of the two-bit memory operation ('01' → '10'). Since the previous operation started from '00', we need to first perform input preparation to change the initial state from '00' to '01'. For this, we apply the pulse input directly to the second bit to convert it from '0' to '1' (see the $\varphi_1$ and $\varphi_3$ profiles in the input preparation process in Fig. 5c). Note that this pulse input to the second bit does not affect the first bit, because $\varphi_1$ is not constrained. That is, $\varphi_1$ and $\varphi_3$ rotate in the same direction at the same rate without flipping the status of the first bit. Now we apply the pulse input to the first bit. Unlike the monotonously increasing pulse input

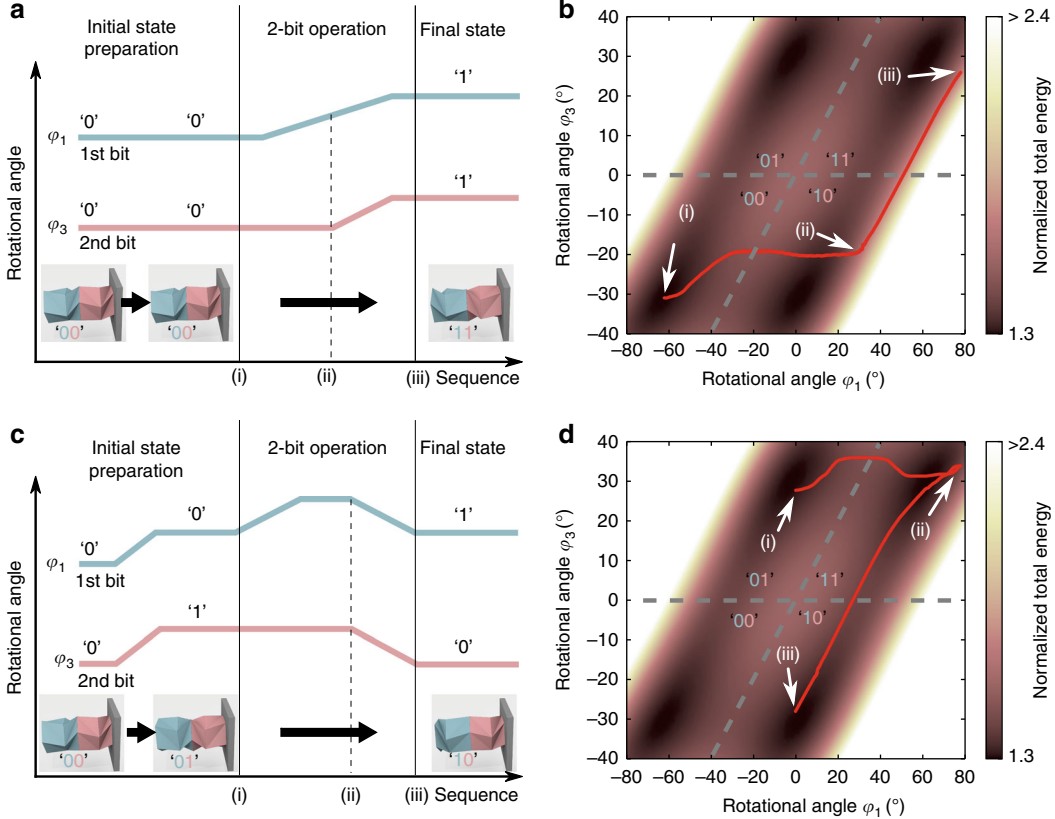

**Fig. 5** Two pairs of TCO cells' to demonstrate two-bit memory operation. **a** The sequence of the operation for '00' → '11' is shown. A pulse input is applied only to the first bit. **b** Experimental result in a *red curve* indicates that the operation flips the second bit from '0' to '1'. **c** The sequence of the operation for '01'→ '10' is shown. A pulse input is applied to the second bit first for the input preparation, and then to the first bit for the operation. **d** Experimental result indicates that the operation flips the second bit from '1' to '0'

needed for the previous operation of '00' → '11', the operation of '01' → '10' requires the increasing—then decreasing—trend of the pulse input (compare the $\varphi_1$ profiles between Fig. 5a, c. See Supplementary Note 9 for the detailed pulse shape). This is due to the intermediary step of '11', which is positioned at the high value of $\varphi_1$ (see Fig. 4b). Up on the application of the pulse input, we initially observe the first bit changes from '0' to '1' as shown by the trajectory of the red curve from the state (i) to the top right corner (ii) in Fig. 5d. Sequentially, as the pulse input decreases, the energy state moves from the state (ii) to the state (iii), flipping the second bit from '1' to '0'. Since distinctive $u_C$ values are applied to the first and second bits, the energy slope from '11' to '10' is less than that from '11' to '01'. Therefore, we observe that the state changes from '11' to '10' instead of '11' to '01' (see Supplementary Note 9 for more details). This serial process eventually changes the combined states of the first and second bits from '01' to '10', successfully verifying the second case of the two-bit memory operation (Supplementary Movie 7).

## Discussion

In this study, we have analytically calculated and experimentally and numerically demonstrated versatile folding motions of volumetric origami, which can feature mono-/bi-stability, zero-stiffness mode, and bifurcation behavior. We have shown that these tunable origami units can be used as non-volatile memory cells by assembling them hierarchically in single-cell, double-cell, up to four-cell levels. Although this study focused on one-dimensional systems, we envision that the origami system can be further extended to multi-dimensions, e.g., honey-comb like 3D clusters. This will function as a layer of mechanical memory

storage and computing structures. Likewise, while this study explored only serial connections of origami cells, they can be also connected in parallel or coaxially, to achieve various functionalities (See Supplementary Note 10 and Supplementary Fig. 11 for conceptual extensions of the origami cells in planar and serial fashions for the potential realization of multi-bit systems. Also see references[30, 31] for similar arrangements or concepts). Moreover, the versatile nature of the TCO together with nanoelectromechanical systems (NEMS) has great potential to develop robust NEMS actuators and sensing devices[7, 32]. In addition, the intrinsic nature of the TCO cells that interweave axial and torsional motions can be further exploited for dynamic purposes, e.g., reusable impact mitigating system. Conclusively, the volumetric origami cells can pave a new way for designing novel engineering systems for mechanical computing and other purposes relying on their rich constitutive mechanics in a single-cell level and strong cohesion in a multi-cell level.

## Methods

**Principle of minimum total potential energy approach.** By using the geometry of the TCO-based unit cell, we calculate the length of crease lines $a$ and $b$ (see Fig. 1c) as follows:

$$a = \sqrt{(h_0 - u)^2 + 4R'^2 \sin^2\left(\frac{\varphi}{2} + \frac{\theta_0}{2} - \frac{\pi}{2n} + \theta_{cal}\right)}$$

$$b = \sqrt{(h_0 - u)^2 + 4R'^2 \sin^2\left(\frac{\varphi}{2} + \frac{\theta_0}{2} + \frac{\pi}{2n} - \theta_{cal}\right)}$$

(1)

where $R'$ and $\theta_{cal}$ are a modified radius of the circle circumscribing the cross-

section and a calibrated angle to compensate for the difference between original and physical prototype models (Please see Supplementary Note 2 for details). Then, the total elastic energy is calculated as $U = \frac{1}{2}nk(a - a_0)^2 + \frac{1}{2}nk(b - b_0)^2$ where $a_0$ and $b_0$ are initial length of $a$ and $b$, and $k$ is a spring constant of the truss members. Also, the work is obtained by $W = Fu + T\varphi$ where $F$ and $T$ are the external force and torque applied to the TCO cell, respectively. Based on these expressions, the total potential energy ($\Pi$) is

$$\Pi(u, \varphi) = U - W = \frac{1}{2}nk(a - a_0)^2 + \frac{1}{2}nk(b - b_0)^2 - Fu - T\varphi \qquad (2)$$

By applying the principle of minimum total potential energy (i.e., $\partial\Pi/\partial u = 0$ and $\partial\Pi/\partial\varphi = 0$)[33], we obtain the analytical expressions of the two-DOF folding/unfolding motion of the TCO-based structure (Supplementary Notes 1 and 2).

**Prototype fabrication and compression test**. We use acrylic plates tailored by a laser cutter for the top and bottom polygons, and 3D printed parts made of polylactic acid for universal joints to attach truss elements to the polygons. Stainless steel shafts (diameter is 3.18 mm) and linear springs ($k = 3.32$ kN m$^{-1}$ for monostable, bistable, zero-stiffness models; $k = 1.08$ kN m$^{-1}$ for bifurcation model) are used to form truss elements that support tension/compression. To obtain the mechanical properties of the prototypes, we build a customized testing setup where the prototype is placed horizontally and its bottom surface is mounted on a fixed wall. The top surface is supported by a ball bearing and stainless steel shaft, so that it can translate and rotate with minimal friction (Supplementary Fig. 2, Supplementary Note 3, and Supplemental Movies 2–7).

**Data availability**. Data supporting the findings of this study are available from the corresponding author on request.

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

## Acknowledgements

We thank Dr. M. Clark at CoMotion at the University of Washington for technical support. We also thank Professor H. Lee at KAIST and Dr. H. Kim at Samsung Advanced Institute of Technology in Korea for helpful discussions. We are grateful for the support from the ONR (N000141410388) and NSF (CAREER-1553202), and the Washington Research Foundation.

## Author contributions

H.Y. and M.L. conducted the research and interpreted the results, and J.Y. and T.T. provided guidance throughout the research. H.Y., T.T., and J.Y. prepared the manuscript.

## Additional information

**Competing interests:** The authors declare no competing financial interests.

