## [Peer Review File · Nature Communications]

Reviewers' comments:

Reviewer #1 (Remarks to the Author):

Title: Origami-based tunable truss structures for non-volatile mechanical memory operation

Authors: H Yasuda, et al.

Summary: The authors present an origami-inspired mechanism that demonstrates a mechanical bistability. The authors then use that bistability to create a torsion-driven device that they discuss as a mechanical "bit." The manuscript demonstrates theoretical modeling in conjunction with experimental data to demonstrate both 1 and 2 bit systems.

I generally think this is an interesting and well-written manuscript with valuable content. I have a few comments that, if addressed, would give me confidence in recommending publication. Specifically:

1) Are there theoretical limits to the number of bits a 1D device can support? How about practical limits? How does the energy barrier height vary with the number of bits? In the end of the manuscript, there's mention of other configurations such as 2D lattices and 3D clusters. This is very appealing as a concept, but given the practical challenges and complexity of an addressable array of input torque device, does this seem too far-fetched? I'm not going to ask for a demonstration as a condition for acceptance since the structures are hand-assembled and the amount labor would be unreasonable. Nevertheless, I would appreciate an honest discussion in the text about what would be involved for these aspirational devices capable of mechanical memory and/or computing.

2) Can computation be performed on/with these devices, or is it simply storage? If so, what would be the basic operation of an "and" or "or" gate? Is there any simple way to mechanically "read" the stored data or are camera-vision based methods generally required? Does reading the data with an applied twist actually alter the data too? If so, then it would appear that the act of reading is "destructive"... What limitations does the practical challenges of reading impose for applications?

3) I think the use of "overconstrained" (pg 2) is confusing and should either be clarified or removed. In one hand, I understand the engineering community uses this term to specifically describe linkage structures with more DOF than is predicted by DOF/constraint-counting arguments (floppy systems). In the other hand, the materials community seems to use the term more to describe systems where there are more constraints than DOF (rigid systems). In other words, it means opposite things to different communities! Since the paper seems to be relevant for both audiences, I would caution against this particular description of the Miura-ori.

4) More discussion of Figure 2 on why the experimental and theoretical curves deviate, particularly for panel 2C, would be helpful. The text asserts the differences are from friction of the joints, but is this based on specific measurements? If there's no specific data supporting the claim, then I think it's a hypothesis (and should be stated as such) until shown otherwise. Comparing the scale of the axes, it looks like the normalized deformation varies by up to 6-fold between structures. Why is there so much variability here? Why do the experimental data probe such different amounts of deformation? Since it's normalized, I would have expected to see a consistent range of strains being probed. And the bifurcation in panel 2D looks like a classic pitchfork bifurcation – is this true? If so, why not state it? If not, then what type of bifurcation is it?

5) Demonstrating coupled flips $00 \rightarrow 11$ and $01 \rightarrow 10$ is nice and generally something I haven't seen before. But what about the more vanilla flips $00 \leftrightarrow 01$, $00 \leftrightarrow 10$, $11 \leftrightarrow 01$, and $11 \leftrightarrow 10$? I see

SFig 8 has indications of individual flips in sequence such as 00 → 10 → 11 and 01 → 11 → 10, but are these individual steps always reversible from the same initial starting configuration? Are these transitions possible to trigger with the same applied torsion as in the movies? Also, I didn't get a clear sense if any of the double-flips are reversible, e.g., 11 → 00 and 10 → 01. I think these questions would be most easily answered with a figure similar to SFig 8A that shows trajectories for all 4 transitions and their inverses. Also, the discussions of "input pulses" toward the end of the manuscript, and how the system was prepared with pre-compressions, could be made clearer to give a better sense of the system's full capabilities and/or limitations.

6) While the analytical results seem reasonable, I did not specifically check the derivations of the equations in the supplemental file.

To reiterate, my assessment of the manuscript, overall, is very positive. I encourage the authors to view this report as constructive feedback and find it helpful in making revisions.

Reviewer #2 (Remarks to the Author):

This paper demonstrates an origami fold pattern that can exhibit bistable (or continuous deformations). The idea of the authors is that the bistable configurations can be used to store information. They then fabricated more complex mechanical structures based on this initial origami design. This could then be extended toward larger structures and even in 2D, though that is not demonstrated here.

I think the fold pattern is interesting, though I admit that I have seen some other patterns like it. I also think their analysis of the one- and two-bit configurations is interesting. There are a few things missing, however, that trouble me. In no particular order:

1. how do they propose to read the memory state? In particular, a useful mechanical memory could be read and stored mechanically. I don't see any mechanism described for reading the memory in this paper.

2. I don't understand how to scale this up. In particular, the description of addressing two linear bits seems a bit intricate (though interesting). But the process seems very specific to two bits. How would three bits be set individually?

The point is not that this isn't interesting, but that you can't claim you have a mechanical memory without being able to answer both questions. In particular, bistable mechanical structures are not hard to find so there has to be more than bistability behind a proper mechanical memory.

My inclination is that if the authors did either demonstrate such a thing or at least explain better how it could be addressed, this would be a great paper. As it is now, it seems incomplete.

Some minor comments:

The last sentence of the caption of Figure 3: "Here, the configuration of the folded right cell represents '0', while the one with the folded right cell denotes '1'." One of those should be the folded left cell, right?

I don't understand the heading: "One-bit memory operation: Two-bit memory operation". Am I

supposed to read the colon as "to"? In any case, this section is not about one-bit memory operation and it might be clearer if you rename this section.

First Reviewer (Reviewer's comments in blue font):

Summary: The authors present an origami-inspired mechanism that demonstrates a mechanical bistability. The authors then use that bistability to create a torsion-driven device that they discuss as a mechanical “bit.” The manuscript demonstrates theoretical modeling in conjunction with experimental data to demonstrate both 1 and 2 bit systems.

I generally think this is an interesting and well-written manuscript with valuable content. I have a few comments that, if addressed, would give me confidence in recommending publication. Specifically:

We thank the Reviewer for helpful comments and suggestions. We also want to point out that the following comments by the Reviewer significantly helped us improve the quality of the manuscript.

1-1) Are there theoretical limits to the number of bits a 1D device can support? How about practical limits? How does the energy barrier height vary with the number of bits?

This is an excellent question. We expect that expanding this origami building block to a multi-bit storage system beyond the two-bit system will be challenging. As demonstrated in this manuscript, even the two-bit system requires a strong coupling mechanism between neighboring cells. A full control over multiple bits beyond two may require close interactions among multiple cells, which would not be easily doable using conventional mechanical memory systems. To the best of the authors’ knowledge, the demonstration of multi-bit systems beyond two-bit has not been thoroughly explored in other realms (e.g., optics, electronics) either due to this reason.

However, we envision that the expansion of our origami architecture to a multi-bit system would be theoretically possible by using the following two approaches:

- (1) **Usage of mechanical connectors:** If we can have control over the connection of each cell (e.g., mechanical clutch that engages and disengages the mechanical connections among origami cells in a controllable way), we can possibly build a multi-bit system in combination of one- and two-bit memory cells. Figure-R1 below explains such possibility. Here we see an example of converting $\langle 000000 \rangle$ to $\langle 101001 \rangle$ by arranging single- and double-bit operations in sequences, which are enabled by turning coupling on and off controllably. The concept of combining single and double bits for multi-bit operations has also been explored in [1].

Figure-R1: Multi-bit operations in the proposed origami architecture enabled by controllable connections between cells. Vertical lines with connection and dis-connection imply engagement and disengagement of cells, respectively.

- (2) **Usage of a graded TCO chain with varying potentials:** In case we do not have a control over the inter-cell connections, an operation like the good-old rotary dial key of a safe can be used. Here,

the 1D device will be composed of a graded chain of TCO unit cells. That is, the chain will be sorted according to the magnitude of the energy barrier, e.g., the left-most has the most flexible architecture (i.e., lowest energy barrier), while the right-most has the stiffest architecture (highest energy barrier). As noted in the main manuscript, such variations of energy barrier can be achieved by imposing different values of pre-compression to identical TCO cells. Given the above example, let $X(i) = 0$ or 1 denotes the i -th bit of the graded system, we start from an initial state:

$$X = \langle 000000 \rangle$$

and want to turn it to the target state:

$$T = \langle 101001 \rangle.$$

Now, if we rotate X in counterclockwise or clockwise direction, then we can turn the bits to $\langle 111111 \rangle$ or $\langle 000000 \rangle$, respectively. However, because of the graded stiffness of TCOs, this happens in a controlled sequential order from the left to the right as:

$$\langle 000000 \rangle$$

$$\langle 100000 \rangle$$

$$\langle 110000 \rangle$$

$$\langle 111000 \rangle$$

$$\langle 111100 \rangle$$

Therefore, we suggest the following algorithm to obtain the desired state T :

```

while ( $X \neq T$ ) {
  1.  $X' \leftarrow X$  ( $X'$  is the "primal" configuration)
  2. Compare the right-most bit of  $T$  and  $X$  that are different, i.e.,  $T(i) = X(i)$  where  $(i = k + 1, \dots, n)$ . If  $T(k) = 1$  or  $0$ , rotate counterclockwise or clockwise, respectively, to obtain the new state  $X$ . Here, because of the higher potential barrier in  $(k + 1)$ -th TCO bit, only first  $k$  bits are initialized, i.e.,
      
$$X(i) = X'(i) \text{ where } i = k + 1, \dots, n$$

$$X(i) = T(k) \text{ where } i = 1, \dots, k$$

}

```

If we apply this algorithm to the example, X changes along:

$$\langle 000000 \rangle$$

$$\langle 111111 \rangle$$

$$\langle 000001 \rangle$$

$$\langle 111001 \rangle$$

$$\langle 001001 \rangle$$

$$\langle 101001 \rangle.$$

Thus, even if we do not have direct control over the inter-cell connections, we can achieve multi-bit operations by using our TCO cells.

In both cases, the number of bits can be limitless ideally, if we disregard this friction or any other dissipative effects. However, the Reviewer raised an important point that there would exist a practical limit in terms of the number of cells connected, because of such dissipative effects.

In the conclusion of the updated manuscript, we have briefly commented on the feasibility of expanding our system to a multi-bit system. We have also added new Supplementary Note 10 to the Supplementary Document to discuss this.

1-2) In the end of the manuscript, there's mention of other configurations such as 2D lattices and 3D clusters. This is very appealing as a concept, but given the practical challenges and complexity of an addressable array of input torque device, does this seem too far-fetched? I'm not going to ask for a demonstration as a condition for acceptance since the structures are hand-assembled and the amount labor would be unreasonable. Nevertheless, I would appreciate an honest discussion in the text about what would be involved for these aspirational devices capable of mechanical memory and/or computing.

Thank you for the comment. We agree with the Reviewer that the expansion of our system to 2D and/or 3D systems will involve more complexity and practical difficulties. Despite such challenges, the authors envision that they would provide enhanced degrees of freedom in controlling the behavior of origami network for mechanical memory and/or computing purposes. For example, planar arrangements of origami can enable tight network of unit cells through rotational motions. See a conceptual illustration in Figure-R2 below, where a twisting motion of a cell will affect the rotations of neighboring cells. In this rotational network, 'on' and 'off' states of a bi-stable cell (i.e., folded and unfolded states of a bi-stable cell) can function as a spin analogous to the behavior of electrons. This can, in turn, activate or deactivate the propagation of rotational motions, or switch between clockwise and counter clockwise interactions among neighboring cells (see exemplary operations in Figure-R2). This can be exploited for realizing coupling and decoupling of neighboring bits. We want to point out that a similar concept of 2D torsional network has been proposed for a wave dynamics application [2]. 3D clusters of origami will combine the axial and planar dimensions of such origami architectures.

We have added a new section in the Supplemental Material (Supplementary Note 10) to propose this planar concept and visualize such architectures. In the last paragraph of the added section, we have also stated the challenges associated with the process of constructing multi-dimensional architectures.

Figure-R2: Planar arrangements of cylindrical origami cells for exemplary operations of (a) '11' \rightarrow '00' and (b) '10' \rightarrow '01', enabling two-bit operations similar to the behavior of the axially arranged origami cells presented in our manuscript. Note that in this example, the spin directions of the two cells are identical. However, we can also change the configurations of their spin directions and/or torque directions to achieve other operations.

2-1) Can computation be performed on/with these devices, or is it simply storage? If so, what would be the basic operation of an “and” or “or” gate?

This is another good point. The proposed origami system is capable of handling two-bit operations, and previous studies have shown that such two-bit operations can be associated with the controlled NOT (cNOT) gate, which is phenomenologically analogous to quantum logic gate. Here, however, we do not claim that our origami system is compatible with the quantum computing systems. This is because our system is not inherited to the coherence concept that quantum computers possess.

Here we briefly describe the operation of the cNOT gate. The cNOT gate is composed of two bits; one is called “control bit,” and the other is “target bit”. The key operation of the cNOT gate is flipping “target bit” if “control bit” is $|1\rangle$. Figure-R3 shows such mechanisms of the cNOT operation. For the first four operations, it shows that if the control bit is $|1\rangle$, it converts the target bit from $|0\rangle$ to $|1\rangle$ or vice versa.

Mathematically, this process determines the state of the control bit in the form of $\alpha|0\rangle + \beta|1\rangle$, where α and β are the coefficients satisfying $\alpha + \beta = 1$. The key point here is to verify that the onset of the control pulse (i.e., $|1\rangle$) can flip the information stored in the target bit. This process can be reduced to the following two cases: $\alpha|00\rangle + \beta|11\rangle$ and $\alpha|01\rangle + \beta|10\rangle$. The former corresponds to the conversion of the target bit from off to on (i.e., $|0\rangle$ to $|1\rangle$), as the control bit is turned on. The latter implies the opposite case that the target bit changes from on to off (i.e., $|1\rangle$ to $|0\rangle$). The last two rows of the Table below show these cNOT operations, and the processes of such operations in our origami settings are shown graphically in the right panel of Figure-R3 in terms of φ_1 and φ_2 . This shows that our system can be extended to perform two-bit operation, specifically cNOT operation using the current setup. The authors suggest the Reviewer to refer to [3], which reports a similar cNOT operation by using a pulse technique similar to the approach taken in this study.

It should be noted in passing here that this cNOT operation is similar to the two-bit memory operation that we have presented in our manuscript. However, we did not mention this cNOT word intentionally, to avoid confusion with the quantum computing concept as stated in the first paragraph above. Nonetheless, it is evident that the two-bit operations are closely associated with the cNOT operation.

Figure-R3: (Left) Table that shows (top four rows) single-bit flipping operations and (bottom two rows) cNOT operations. (Right) Time history of φ_1 and φ_2 for demonstrating the cNOT operation.

2-2) Is there any simple way to mechanically “read” the stored data or are camera-vision based methods generally required? Does reading the data with an applied twist actually alter the data too? If so, then it would appear that the act of reading is “destructive”.... What limitations does the practical challenges of reading impose for applications?

We appreciate the Reviewer’s comment. In this study, we have used a pair of laser Doppler vibrometers to determine the two-bit states by measuring the rotational angles φ_1 and φ_3 individually. Given the two-bit configuration of our system, this optical technique will be a straightforward method to read the data stored in each cell of the origami system. Note that previous studies have also reported the usage of optical methods (e.g., optical waveguides) for mechanical memory readout for a single-bit system [4, 5]. We also expect that a capacitance-based method (e.g., [6]) would work efficiently, when the polygons (i.e., facesheets) of the proposed origami-based system are made of electrically conductive materials.

In addition to the proposed method based on the laser vibrometry, here we would like to add another “mechanical” method that can be used for reading the data from our origami system. The challenges associated with reading the two-bit memory is that we need to read not only the control bit information (which is adjacent to the actuation side), but also the target bit (which is apart from the actuation side) at the same time. To readout both bits simultaneously, we propose to apply rotational perturbations (φ_1) to the actuation side and measure the torque profile as a function of φ_1 . Please note that we only control the end of the system (the left-most cross-section in the insets of Figure-R4) to extract bit information. For example, we consider the single-bit case of $(h_0, \theta_0) = (90 \text{ mm}, \pm 46^\circ)$ and the pre-compression of $u_C = 45 \text{ mm}$. In this configuration, there are two local minimum energy states ($\varphi_1 = -29^\circ$ for ‘0’ and $\varphi_1 = +29^\circ$ for ‘1’). Then, we apply angular perturbation of $\Delta\varphi_1 = \pm 15^\circ$ and measure torque at the left-most polygon (i.e., the torque is measured in the range of $\varphi_1 = -29^\circ \pm 15^\circ$ for ‘0’ case, and $\varphi_1 = +29^\circ \pm 15^\circ$ for ‘1’). Please note that $\Delta\varphi_1 = \pm 15^\circ$ is chosen not to overcome its energy barrier in this angular range, so that it avoids destructing bit information. Figure-R4a shows the result of numerical calculations for the single-bit case, and we obtain two distinctive torque curves, which indicate that bit information can be extracted based on the application of rotational angle variation ($\Delta\varphi_1$).

We can also extend this readout method to the two-bit case. Here, we consider the case of $(h_0, \theta_0) = (90 \text{ mm}, [+46^\circ, -46^\circ, +46^\circ, -46^\circ])$ and the pre-compression of $u_C = 50 \text{ mm}$ and 47.5 mm for the first and second bit (the same condition as the one described in the main manuscript). Now we apply to the leftmost polygon the rotational perturbations of $\Delta\varphi_1 = \pm 15^\circ$ with respect to the four local minimum energy states: $(\varphi_1, \varphi_3) = (-63^\circ, -31^\circ)$ for ‘00’; $(+2^\circ, -31^\circ)$ for ‘10’; $(-2^\circ, +31^\circ)$ for ‘01’; and $(+63^\circ, +31^\circ)$ for ‘11’. Figure-R4b shows our calculation results, where we observe four different curves for each bit state. Therefore, we could determine the bit state by applying $\Delta\varphi_1$ and measure the torque profile. Alternatively, the frequency response of the system can be also recorded to predict the torsional stiffness of the system and thereby to read its memory state. We want to point out that because ‘01’ and ‘10’ cases show similar torque curves, the practical challenge would be a precise measurement of the torque under external influence such as noise and friction. We expect that these torque profiles can be further separated by changing u_C values or the initial twist angles imposed on both cells.

Figure-R4: Read out bit information for single-bit and two-bit systems. (a) We consider a single-bit system with $(h_0, \theta_0) = (90 \text{ mm}, \pm 46^\circ)$, and we apply perturbation of rotational angle ($\Delta\varphi_1$) around each local minimum energy state ($\varphi_1 = -29^\circ$ for '0' and $\varphi_1 = +29^\circ$ for '1'). The blue and red curves show torque change around '0' and '1', respectively. (b) The same perturbation of rotational angle is applied to a two-bit system with $(h_0, \theta_0) = (90 \text{ mm}, [+46^\circ, -46^\circ, +46^\circ, -46^\circ])$, and we obtain torque change. The blue, green, black, and red curves indicate torque change around '00', '10', '01', and '11' respectively.

3) I think the use of “overconstrained” (pg 2) is confusing and should either be clarified or removed. In one hand, I understand the engineering community uses this term to specifically describe linkage structures with more DOF than is predicted by DOF/constraint-counting arguments (floppy systems). In the other hand, the materials community seems to use the term more to describe systems where there are more constraints than DOF (rigid systems). In other words, it means opposite things to different communities! Since the paper seems to be relevant for both audiences, I would caution against this particular description of the Miura-ori.

We appreciate the Reviewer's suggestion. In the updated manuscript, the term “overconstrained” was removed to avoid the confusion.

4-1) More discussion of Figure 2 on why the experimental and theoretical curves deviate, particularly for panel 2C, would be helpful. The text asserts the differences are from friction of the joints, but is this based on specific measurements? If there's no specific data supporting the claim, then I think it's a hypothesis (and should be stated as such) until shown otherwise.

We thank the Reviewer for the comment. Figure-R5 below shows displacement vs. normalized energy graphs for monostable, bistable, and zero-stiffness modes plotted all together. Here, solid curves represent analytical data, while dashed curves with bands denote experimental data with standard deviations. First, we observe that the experimentally measured energy in the zero-stiffness mode is extremely small compared to the energy levels by other two cases (see the blue curves in Figure-R5). This is because the zero-stiffness mode picks up energy slowly as noted in the manuscript. In this normalized energy scale as shown in Figure-R5, the discrepancy between the analytical and experimental data is not very noticeable.

To measure this extremely low energy level, the authors have spent a significant amount of time and effort during experiments. In the early stage of testing, we observed a relatively high level of energy. However, as we doped lubrication agents in mechanical hinges (e.g., WD-40), we were able to reduce the

energy level significantly. Though we have not included such comparison data for the sake of simplicity, this confirms the effect of friction.

While we were able to verify the frictional effect, the authors agree that this is still hypothetical, since there might be other factors that contribute to the mismatch between the analytical and experimental results. Thus, we have modified the related sentence as below:

(Before change) “The discrepancy between the analytical and experimental results attributes to the friction of the mechanical joints in the truss elements.”

(After change) “The discrepancy between the analytical and experimental results may attribute to the dissipative factors, including the friction of the mechanical joints in the truss elements.”

We have also briefly mentioned such effects of friction in Supplementary Note 3 in the updated manuscript.

Figure-R5: Normalized energy stored in origami as a function of normalized displacements. Solid curves represent analytical data, while dashed curves denote experimental data with errors.

4-2) Comparing the scale of the axes, it looks like the normalized deformation varies by up to 6-fold between structures. Why is there so much variability here? Why do the experimental data probe such different amounts of deformation? Since it’s normalized, I would have expected to see a consistent range of strains being probed.

In the caption of Figure 2 of the main manuscript, the authors have stated that “In the experimental curves, the range of u/h_0 is restricted by the folding motions of the TCO-based truss prototypes. For example, the highly twisted shape of the zero-stiffness TCO prototype (Fig. 1g) causes the truss elements overlap in the early stage of folding, allowing only $\sim 15\%$ of u/h_0 as shown in the panel (c). The moderately twisted geometry of the monostable and bistable cases (Fig. 1e and f) permit more

compression, allowing approximately 50% folding of the truss structure in terms of u/h_0 as shown in the panels (a) and (b).”

To clarify this point, we have added the following sentences in the main manuscript, along with a new *Supplementary Movie 3*.

“The experimental measurements with standard deviations are denoted by dashed curves with bands. Note that in experiments, the range of u/h_0 is restricted by the folding motions of the TCO-based truss prototypes (*Supplementary Movie 3*). Within the measurement range, the experimental data corroborate these analytical results.”

4-3) And the bifurcation in panel 2D looks like a classic pitchfork bifurcation – is this true? If so, why not state it? If not, then what type of bifurcation is it?

We agree with the Reviewer’s comment. We modified the sentence accordingly:
“The two different trends in the uni-axial testing verify this *pitchfork* bifurcation behavior”

5-1) Demonstrating coupled flips $00 \rightarrow 11$ and $01 \rightarrow 10$ is nice and generally something I haven’t seen before. But what about the more vanilla flips $00 \leftrightarrow 01$, $00 \leftrightarrow 10$, $11 \leftrightarrow 01$, and $11 \leftrightarrow 10$?

The key functionality of two-bit memory is a bit-flipping behavior. That is, the onset of the control bit will cause the flipping of the target bit. Such bit-flipping behavior has been demonstrated by ‘00’ \rightarrow ‘11’ and ‘01’ \rightarrow ‘10’ in this manuscript. The vanilla flips that the Reviewer has brought up (e.g., ‘00’ \leftrightarrow ‘01’, ‘00’ \leftrightarrow ‘10’, ‘11’ \leftrightarrow ‘01’, and ‘11’ \leftrightarrow ‘10’) are basically one-bit operations. Thus, they would be achievable if we control each origami cell individually, or if we change the width of the trapezoidal signal (e.g., ‘00’ \rightarrow ‘10’ is achievable if we apply half of the input pulse that we used in the manuscript. Further explanation to follow in the next paragraph). For the combined two-bit operation, however, they would not comply with the bit-flipping mechanism realized through this study.

Here we would like to provide the further analysis results to explain what operations are possible and not possible in the current origami settings. Figure-R6 shows our experimental and computational results of two-bit operation. In this figure, blue and green curves indicate stable and unstable paths of the two-bit system, respectively, based on the calculation of $\partial U / \partial \varphi_3 = 0$. The red curves have been obtained from the experiments. For the operation of ‘00’ \rightarrow ‘11’ as shown in Figure-R6a, once the system departs from ‘00’, it quickly jumps to ‘10’ to avoid the unstable branch. After that, it continues following the stable branch to reach ‘11’ state. As mentioned in the previous paragraph, if the range of φ_1 is restricted to near zero angle, the operation will stop in the state of ‘10’, achieving ‘00’ \rightarrow ‘10’.

For the operation of ‘01’ \rightarrow ‘10’ as shown in Figure-R6b, the system initially changes its bit state from ‘01’ to ‘11’ by skipping the unstable branch. Once the control signal changes its direction (i.e., φ_1 starts to decrease), the system follows the stable branch that connects ‘11’ and ‘10’. Here, we also see that the operation of ‘01’ \rightarrow ‘11’ is possible if the input signal in terms of φ_1 stops at its peak point near 60°.

In passing here, we want to note that given the trapezoidal signal described in the manuscript and Supplemental Figure 9, we expect that our system exhibits the two-bit operations. However, if we alter the shape or range of the input signal, we will be able to achieve other operations as stated above, e.g., ‘00’ \rightarrow ‘10’ and ‘01’ \rightarrow ‘11’. In the updated manuscript, we have expanded the “*Supplementary Note 9: Two-bit operation*” to include explanations on the trajectories of (φ_1, φ_3) based on our stability analysis.

Figure-R6: (a) Two-bit operation for ‘00’ \rightarrow ‘11’. The normalized energy of the two-bit system as a function of φ_1 and φ_3 is shown. The blue and green curves are obtained from $\partial U/\partial \varphi_3 = 0$, and blue and green lines indicate stable and unstable paths, respectively. Experimental results are shown in the red curves. Red circles are the states where the system takes local minimum energy. (b) Experimental result of two-bit operation for ‘01’ \rightarrow ‘10’ is shown.

5-2) I see SFig 8 has indications of individual flips in sequence such as 00 \rightarrow 10 \rightarrow 11 and 01 \rightarrow 11 \rightarrow 10, but are these individual steps always reversible from the same initial starting configuration? Are these transitions possible to trigger with the same applied torsion as in the movies?

This is an excellent question. The operation of ‘01’ \rightarrow ‘11’ \rightarrow ‘10’ will be reversible (see Figure-R6b), but the operation of ‘00’ \rightarrow ‘10’ \rightarrow ‘11’ would not be reversible (Figure-R6a). This is because in Figure-R6a, the system will jump from ‘11’ to ‘01’, instead of following the trajectory of ‘11’ to ‘10’. This is to avoid the unstable branch as described in the response of the comment 5-1. Please note that the nature of reversibility depends on the type of input function (i.e., the waveform of the trapezoidal function in this study). We also want to point out that for the fulfillment of the two-bit operation, the reversibility is not necessary, since ‘11’ \rightarrow ‘00’ and ‘10’ \rightarrow ‘01’ would violate the ‘bit-flipping’ functionality of the proposed system. For the bit flipping, the target bit would flip only when the control bit is on (i.e., ‘1’).

Regarding the torque to trigger a transition, the applied torque will be different depending on the bit state. New Supplementary Figure 8 (or Figure-R4 included in the response to the comment 2-2) shows relationships between torque and the rotational angle variation ($\Delta\varphi_1$). For example, ‘00’ \rightarrow ‘10’ and ‘10’ \rightarrow ‘11’ cases correspond to the blue and green lines in the positive region of $\Delta\varphi_1$. We witness that as we increase $\Delta\varphi_1$, these two torque curves deviate from each other.

5-3) Also, I didn’t get a clear sense if any of the double-flips are reversible, e.g., 11 \rightarrow 00 and 10 \rightarrow 01. I think these questions would be most easily answered with a figure similar to SFig 8A that shows trajectories for all 4 transitions and their inverses.

This is another good point. In our current study, we focus on one-way control of the target bit (i.e., control bit manipulates the target bit flipping, but not the other way around). This necessitates irreversibility as stated in the response to the previous comment. For example, as demonstrated in this study, ‘01’ \rightarrow ‘10’ is achievable, since the onset of the control bit changes the bit information of the target bit. However, the reverse operation of ‘01’ \rightarrow ‘10’ is not supported, since the change of control bit from 1 to 0 will not incur the change of the target bit. In passing here, we want to note that our system is versatile enough to achieve ‘11’ \rightarrow ‘00’ and ‘10’ \rightarrow ‘01’. But this does not mean the process is reversible, since the trajectory

(i.e., intermediary step) may be different from that of ‘00’ → ‘11’ and ‘01’ → ‘10’. Again, we may also need to impose different trapezoidal input signal and/or the amount of pre-compression u_c to the system.

5-3) Also, the discussions of “input pulses” toward the end of the manuscript, and how the system was prepared with pre-compressions, could be made clearer to give a better sense of the system’s full capabilities and/or limitations.

We thank the Reviewer for the suggestion. For the input pulse used in two-bit operations, we added additional explanations to the appropriate paragraphs of the updated manuscript and supplementary document. Also, we have modified Supplementary Figure 6 to describe how we applied pre-compression to the prototype as below.

Modified Supplementary Figure 6: TCO-based mechanical memory storage.

Second Reviewer (Reviewer's comments in blue font):

This paper demonstrates an origami fold pattern that can exhibit bistable (or continuous deformations). The idea of the authors is that the bistable configurations can be used to store information. They then fabricated more complex mechanical structures based on this initial origami design. This could then be extended toward larger structures and even in 2D, though that is not demonstrated here.

I think the fold pattern is interesting, though I admit that I have seen some other patterns like it. I also think their analysis of the one- and two-bit configurations is interesting. There are a few things missing, however, that trouble me. In no particular order:

We thank the Reviewer for helpful comments and suggestions. We also want to point out that the following comments by the Reviewer significantly helped us improve the quality of the manuscript.

1. how do they propose to read the memory state? In particular, a useful mechanical memory could be read and stored mechanically. I don't see any mechanism described for reading the memory in this paper.

(We find this comment is identical to the comment 2-2 by the first reviewer. Thus, we have used the same response as that used for the comment 2-2 by the first reviewer).

We appreciate the Reviewer's comment. In this study, we have used a pair of laser Doppler vibrometers to determine the two-bit states by measuring the rotational angles φ_1 and φ_3 individually. Given the two-bit configuration of our system, this optical technique will be a straightforward method to read the data stored in each cell of the origami system. Note that previous studies have also reported the usage of optical methods (e.g., optical waveguides) for mechanical memory readout for a single-bit system [4, 5]. We also expect that a capacitance-based method (e.g., [6]) would work efficiently, when the polygons (i.e., facesheets) of the proposed origami-based system are made of electrically conductive materials.

In addition to the proposed method based on the laser vibrometry, here we would like to add another "mechanical" method that can be used for reading the data from our origami system. The challenges associated with reading the two-bit memory is that we need to read not only the control bit information (which is adjacent to the actuation side), but also the target bit (which is apart from the actuation side) at the same time. To readout both bits simultaneously, we propose to apply rotational perturbations (φ_1) to the actuation side and measure the torque profile as a function of φ_1 . Please note that we only control the end of the system (the left-most cross-section in the insets of Figure-R4) to extract bit information. For example, we consider the single-bit case of $(h_0, \theta_0) = (90 \text{ mm}, \pm 46^\circ)$ and the pre-compression of $u_C = 45 \text{ mm}$. In this configuration, there are two local minimum energy states ($\varphi_1 = -29^\circ$ for '0' and $\varphi_1 = +29^\circ$ for '1'). Then, we apply angular perturbation of $\Delta\varphi_1 = \pm 15^\circ$ and measure torque at the left-most polygon (i.e., the torque is measured in the range of $\varphi_1 = -29^\circ \pm 15^\circ$ for '0' case, and $\varphi_1 = +29^\circ \pm 15^\circ$ for '1'). Please note that $\Delta\varphi_1 = \pm 15^\circ$ is chosen not to overcome its energy barrier in this angular range, so that it avoids destructing bit information. Figure-R4a shows the result of numerical calculations for the single-bit case, and we obtain two distinctive torque curves, which indicate that bit information can be extracted based on the application of rotational angle variation ($\Delta\varphi_1$).

We can also extend this readout method to the two-bit case. Here, we consider the case of $(h_0, \theta_0) = (90 \text{ mm}, [+46^\circ, -46^\circ, +46^\circ, -46^\circ])$ and the pre-compression of $u_C = 50 \text{ mm}$ and 47.5 mm for the first and second bit (the same condition as the one described in the main manuscript). Now we apply to the leftmost polygon the rotational perturbations of $\Delta\varphi_1 = \pm 15^\circ$ with respect to the four local minimum energy

states: $(\varphi_1, \varphi_3) = (-63^\circ, -31^\circ)$ for '00'; $(+2^\circ, -31^\circ)$ for '10'; $(-2^\circ, +31^\circ)$ for '01'; and $(+63^\circ, +31^\circ)$ for '11'. Figure-R4b shows our calculation results, where we observe four different curves for each bit state. Therefore, we could determine the bit state by applying $\Delta\varphi_1$ and measure the torque profile. Alternatively, the frequency response of the system can be also recorded to predict the torsional stiffness of the system and thereby to read its memory state. We want to point out that because '01' and '10' cases show similar torque curves, the practical challenge would be a precise measurement of the torque under external influence such as noise and friction. We expect that these torque profiles can be further separated by changing u_C values or the initial twist angles imposed on both cells.

Figure-R4: Read out bit information for single-bit and two-bit systems. (a) We consider a single-bit system with $(h_0, \theta_0) = (90 \text{ mm}, \pm 46^\circ)$, and we apply perturbation of rotational angle ($\Delta\varphi_1$) around each local minimum energy state ($\varphi_1 = -29^\circ$ for '0' and $\varphi_1 = +29^\circ$ for '1'). The blue and red curves show torque change around '0' and '1', respectively. (b) The same perturbation of rotational angle is applied to a two-bit system with $(h_0, \theta_0) = (90 \text{ mm}, [+46^\circ, -46^\circ, +46^\circ, -46^\circ])$, and we obtain torque change. The blue, green, black, and red curves indicate torque change around '00', '10', '01', and '11' respectively.

2. I don't understand how to scale this up. In particular, the description of addressing two linear bits seems a bit intricate (though interesting). But the process seems very specific to two bits. How would three bits be set individually?

(We also find that this comment overlaps with the comment 1-1 by the first reviewer.)

This is an excellent question. We expect that expanding this origami building block to a multi-bit storage system beyond the two-bit system will be challenging. As demonstrated in this manuscript, even the two-bit system requires a strong coupling mechanism between neighboring cells. A full control over multiple bits beyond two may require close interactions among multiple cells, which would not be easily doable using conventional mechanical memory systems. To the best of the authors' knowledge, the demonstration of multi-bit systems beyond two-bit has not been thoroughly explored in other realms (e.g., optics, electronics) either due to this reason.

However, we envision that the expansion of our origami architecture to a multi-bit system would be theoretically possible by using the following two approaches:

- (1) **Usage of mechanical connectors:** If we can have control over the connection of each cell (e.g., mechanical clutch that engages and disengages the mechanical connections among origami cells in a controllable way), we can possibly build a multi-bit system in combination of one- and two-bit memory cells. Figure-R1 below explains such possibility. Here we see an example of

converting $\langle 000000 \rangle$ to $\langle 101001 \rangle$ by arranging single- and double-bit operations in sequences, which are enabled by turning coupling on and off controllably. The concept of combining single and double bits for multi-bit operations has also been explored in [1].

Figure-R1: Multi-bit operations in the proposed origami architecture enabled by controllable connections between cells. Vertical lines with connection and dis-connection imply engagement and disengagement of cells, respectively.

- (2) **Usage of a graded TCO chain with varying potentials:** In case we do not have a control over the inter-cell connections, an operation like the good-old rotary dial key of a safe can be used. Here, the 1D device will be composed of a graded chain of TCO unit cells. That is, the chain will be sorted according to the magnitude of the energy barrier, e.g., the left-most has the most flexible architecture (i.e., lowest energy barrier), while the right-most has the stiffest architecture (highest energy barrier). As noted in the main manuscript, such variations of energy barrier can be achieved by imposing different values of pre-compression to identical TCO cells. Given the above example, let $X(i) = 0$ or 1 denotes the i -th bit of the graded system, we start from an initial state:

$$X = \langle 000000 \rangle$$

and want to turn it to the target state:

$$T = \langle 101001 \rangle.$$

Now, if we rotate X in counterclockwise or clockwise direction, then we can turn the bits to $\langle 111111 \rangle$ or $\langle 000000 \rangle$, respectively. However, because of the graded stiffness of TCOs, this happens in a controlled sequential order from the left to the right as:

$$\langle 000000 \rangle$$

$$\langle 100000 \rangle$$

$$\langle 110000 \rangle$$

$$\langle 111000 \rangle$$

$$\langle 111100 \rangle$$

Therefore, we suggest the following algorithm to obtain the desired state T :

while ($X \neq T$) {

1. $X' \leftarrow X$ (X' is the "primal" configuration)

2. Compare the right-most bit of T and X that are different, i.e., $T(i) = X(i)$ where $(i = k + 1, \dots, n)$. If $T(k) = 1$ or 0 , rotate counterclockwise or clockwise, respectively, to obtain the new state X . Here, because of the higher potential barrier in $(k + 1)$ -th TCO bit, only first k bits are initialized, i.e.,

$$X(i) = X'(i) \quad \text{where } i = k + 1, \dots, n$$

$$X(i) = T(k) \quad \text{where } i = 1, \dots, k$$

}

If we apply this algorithm to the example, X changes along:

<000000>
<111111>
<000001>
<111001>
<001001>
<101001>

Thus, even if we do not have direct control over the inter-cell connections, we can achieve multi-bit operations by using our TCO cells.

In the conclusion of the updated manuscript, we have briefly commented on the feasibility of expanding our system to a multi-bit system. We have also added new Supplementary Note 10 to the Supplementary Document to discuss this.

The Reviewer may also want to look at the response to the comment 1-2 raised by the first reviewer. In the response to that comment, we tried to describe how we can make the system expanded in horizontal directions, instead of the vertical setting described in the manuscript.

The point is not that this isn't interesting, but that you can't claim you have a mechanical memory without being able to answer both questions. In particular, bistable mechanical structures are not hard to find so there has to be more than bistability behind a proper mechanical memory.

We appreciate the Reviewer's comments, and we also agree that there are other structures with bistable behaviors such as a buckled beam. The unique feature of our system is the coupling between two neighboring cells, which enabled the directional control of the target bit by using the control bit. We believe this coupling behavior is the key feature beyond the bistability, which enables the proof-of-concept demonstration of two-bit memory in the setting of origami architectures.

My inclination is that if the authors did either demonstrate such a thing or at least explain better how it could be addressed, this would be a great paper. As it is now, it seems incomplete.

We hope that the revisions made in the updated manuscript, including the reading of the mechanical memory and the potential extension of this two-bit to multiple bit systems, are satisfactory.

Some minor comments:

The last sentence of the caption of Figure 3: "Here, the configuration of the folded right cell represents '0', while the one with the folded right cell denotes '1'." One of those should be the folded left cell, right?

We appreciate that the Second Reviewer pointed out this typo. We modified the figure caption in the updated manuscript as follows:

"Here, the configuration of the folded right cell represents '0', while the one with the folded left cell denotes '1' "

I don't understand the heading: "One-bit memory operation: Two-bit memory operation". Am I supposed to read the colon as "to"? In any case, this section is not about one-bit memory operation and it might be clearer if you rename this section.

We appreciate that the Second Reviewer pointed out this. We changed the heading to “Two-bit memory operation” in the update manuscript.

References

1. Rieffel, E. G., & Polak, W. H. (2011). *Quantum computing: A gentle introduction*. MIT Press.
2. R. K. Pal, M. Schaeffer, and M. Ruzzene, Helical edge states and topological phase transitions in phononic systems using bi-layered lattices, *Journal of Applied Physics* 119, 084305, 2016.
3. T. Yamamoto, Y. A. Pashkin, O. Astafiev, Y. Nakamura, J. S. Tsai, Demonstration of conditional gate operation using superconducting charge qubits. *Nature*, 425(6961), 941–944, 2003.
4. Bagheri, M., Poot, M., Li, M., Pernice, W. P. H., & Tang, H. X., Dynamic manipulation of nanomechanical resonators in the high-amplitude regime and non-volatile mechanical memory operation. *Nat Nano*, 6(11), 726–732, 2011.
5. S. Evoy, D. W. Carr, L. Sekaric, A. Olkhovets, J. M. Parpia, and H. G. Craighead, *J. Appl. Phys.* 86, 6072, 1999.
6. D. Roodenburg, J. W. Spronck, H. S. J. van der Zant, and W. J. Venstra, Buckling beam micromechanical memory with on-chip readout, *Appl. Phys. Lett.* 94, 183501, 2009

REVIEWERS' COMMENTS:

Reviewer #1 (Remarks to the Author):

I thank the authors for their time and effort in thoroughly addressing my questions. I feel confident in recommending this manuscript for publication.

Two quick comments I'd like to add from my re-reading of the manuscript:

1) In the abstract and introduction the authors state "Origami has recently received significant interest from the scientific community as a building block for constructing metamaterials," and "Origami has been a popular building block to construct mechanical metamaterials." My concern is that these appear to be misstatements. Origami is not "a building block." Origami is a strategy/method for designing/constructing building blocks. I believe the subtle difference in wording is crucial for accuracy.

2) I noticed the line "The discrepancy between the analytical and experimental results may attribute to the dissipative factors, including the friction of the mechanical joints in the truss elements" likely has a typo. I believe the authors meant "...may be attributed..."

Reviewer #2 (Remarks to the Author):

The authors did a lot of work to satisfy my concerns and I am satisfied that this should be published.

First Reviewer (Reviewer's comments in blue font):

I thank the authors for their time and effort in thoroughly addressing my questions. I feel confident in recommending this manuscript for publication.

We thank the Reviewer for recommending our manuscript for publication.

Two quick comments I'd like to add from my re-reading of the manuscript:

1) In the abstract and introduction the authors state "Origami has recently received significant interest from the scientific community as a building block for constructing metamaterials," and "Origami has been a popular building block to construct mechanical metamaterials." My concern is that these appear to be misstatements. Origami is not "a building block." Origami is a _strategy/method_ for _designing/constructing_ building blocks. I believe the subtle difference in wording is crucial for accuracy.

We appreciate the Reviewer's suggestion. We modified our expressions in our updated manuscript as follows:

In the abstract: **"a method for designing building blocks to construct metamaterials"**

In the introduction: "Origami has been a popular method for designing building blocks to construct mechanical metamaterials"

2) I noticed the line "The discrepancy between the analytical and experimental results may attribute to the dissipative factors, including the friction of the mechanical joints in the truss elements" likely has a typo. I believe the authors meant "...may be attributed..."

We appreciate that the Reviewer pointed out this typo. We modified the figure caption in the updated manuscript and Supplementary Information as follows:

"The discrepancy between the analytical and experimental results may be attributed to the dissipative factors"

Second Reviewer (Reviewer's comments in blue font):

The authors did a lot of work to satisfy my concerns and I am satisfied that this should be published.

We thank the Reviewer for supporting our manuscript for publication.